# <REDACTED>: APPLYING THE LESS-IS-MORE PRINCIPLE TO MEDIA BIAS DETECTION IN NEWS ARTICLES

## ABSTRACT

As misinformation proliferates across news platforms, the need to detect bias, both overt and latent, becomes critical for trustworthy media analysis. Unlike falsehoods, bias often persists in otherwise factually accurate reporting, requiring more nuanced models to detect patterns in framing, source selection, and agenda setting. Leveraging the advanced analytical capabilities of modern Large Language Models (LLMs), we propose a novel approach that combines reasoning mechanisms with bias detection frameworks to create more transparent and objective news content analysis. Our methodology employs a model consensus strategy with multiple reasoning-capable LLMs (Claude 3.7, DeepSeek-R1, o3-mini, and Gemini 2.5) to generate a curated dataset derived from the MN-DS news corpus. This consensus-driven approach ensures robust bias identification across various news categories while maintaining balanced representation. We then fine-tune the Qwen3 4B model on this dataset using Parameter-Efficient Fine-Tuning (PEFT) with Quantized Low-Rank Adaptation (QLoRA) techniques. Using a distance-based coherence scoring algorithm, we demonstrate that smaller models can effectively acquire reasoning and bias detection capabilities when trained on high-quality examples, as evidenced by a 6.3% increase in accuracy compared to the baseline Qwen3 32B. Our findings support the "Less-Is-More" hypothesis for reasoning (LIMO), suggesting that sophisticated bias analysis can emerge without reinforcement learning when models are exposed to well-structured demonstrations. This work contributes to the advancement of ethical journalism by providing a transparent, open-source framework for bias detection in news articles.

## 1 INTRODUCTION

The proliferation of digital news media has exponentially increased information consumption, highlighting the critical need for transparent and unbiased journalism. News articles inherently contain various forms of bias that can significantly influence public opinion and decision-making (Spinde et al., 2021). Despite the journalistic ideal of objectivity, complete neutrality remains elusive due to the inherent subjectivity in language, framing, and topic selection (Hamborg et al., 2019).

The detection and mitigation of bias in news content serve several crucial functions: (1) enhancing readers' awareness of potential slants in reporting, (2) supporting journalists in producing more balanced content, and (3) promoting a more informed democratic discourse by ensuring access to less skewed information (Raza et al., 2022). However, existing approaches to bias detection often rely on simplistic keyword analysis or require extensive labeled datasets that are costly to produce and may themselves contain biases.

Our research leverages recent advancements in Large Language Models (LLMs), which have demonstrated remarkable capabilities in text analysis and classification tasks through few-shot learning paradigms (Brown et al., 2020). Particularly promising are the enhanced reasoning abilities of modern LLMs, enabling them to engage in step-by-step analytical processes before producing outputs (OpenAI et al., 2024; DeepSeek-AI et al., 2025).

We propose a novel approach that leverages the reasoning capabilities of LLMs for sophisticated text analysis in the context of bias detection. Our methodology employs a three-stage process:

First, we develop a comprehensive taxonomy of media bias by integrating classifications from All-Sides (Mastrine et al., 2019), the multi-dimensional framework by Rodrigo-Ginés et al. (2024), and other established sources (Spinde et al., 2021; Raza et al., 2022). This integrated taxonomy enables standardized identification of 18 diverse bias types across news content (see Appendix A for the complete taxonomy).

Second, we implement a model consensus strategy to generate a high-quality dataset. Starting with the MN-DS multilabeled news dataset (Petukhova & Fachada, 2023), we extract a balanced subset of 1,220 articles across various categories and subcategories. Four reasoning-capable LLMs—Claude 3.7 (Anthropic Team, 2025), DeepSeek-R1 (DeepSeek-AI et al., 2025), o3-mini (OpenAI, 2025), and Gemini 2.5 (Google Team, 2025)—analyze these articles using identical prompts to identify and classify biases at four distinct levels of granularity. This structured output is captured in JSON format, allowing for systematic comparison and validation. We employ a distance-based consensus mechanism with Claude 3.7 as our baseline, establishing a threshold that ensures only high-agreement data points are included in our final dataset.

Third, we fine-tune the Qwen3 4B model (Yang et al., 2025) using Parameter-Efficient Fine-Tuning (PEFT) techniques, specifically Low-Rank Adaptation (LoRA) (Han et al., 2024; Hu et al., 2021; Dettmers et al., 2023). This approach enables us to efficiently transfer bias detection capabilities to a smaller model while maintaining high performance. Our methodology builds on the "Less-Is-More" reasoning hypothesis proposed by Ye et al. (2025), which suggests that sophisticated reasoning capabilities can emerge from minimal but well-structured demonstrations without reinforcement learning, challenging the conventional approach of using reinforcement learning (RL) for enhancing reasoning in LLMs.

Our model is fine-tuned to output the complete reasoning process, providing transparency into how it arrives at bias classifications. By explicitly documenting each step of analysis, from identifying linguistic patterns to evaluating framing choices, the model offers its users insight into the decision-making process rather than merely presenting conclusions. This transparency serves multiple purposes: It allows users to understand the specific elements that contribute to bias detection, enables verification of the model's reasoning, and provides educational value by demonstrating systematic bias analysis. The explicit reasoning output also helps mitigate the "black-box" problem that is common in AI systems, fostering greater trust in the model's assessments while encouraging users to improve their own critical evaluation skills when consuming news media.

This research contributes to the fields of computational linguistics and automated media analysis by demonstrating how reasoning capabilities can be effectively applied to the complex task of bias detection, potentially transforming how we evaluate and consume news media. By creating a model that provides structured and transparent analysis of bias in news articles, our goal is to enhance journalistic integrity and readers' critical awareness of media bias in an increasingly complex information landscape.

## 2 METHODOLOGY

Our methodology comprises two main phases: dataset engineering with consensus-based validation and model fine-tuning using Parameter-Efficient Fine-Tuning techniques, as illustrated in Figure 1. This approach ensures high-quality training data while maintaining computational efficiency in the model development process.

### 2.1 DATASET ENGINEERING AND CONSENSUS MECHANISM

We initiated our data collection process with 2,320 articles from the MN-DS multilabeled news dataset (Petukhova & Fachada, 2023), obtained through two collection rounds: an initial batch of 1,500 articles followed by an additional 820 articles to reach our target dataset size. Our sampling strategy prioritized balanced representation across news categories by grouping articles according to their category and subcategory fields, then sampling proportionally from each group to maintain categorical diversity. Articles were filtered to include only those with more than 400 words to ensure sufficient content for bias analysis.

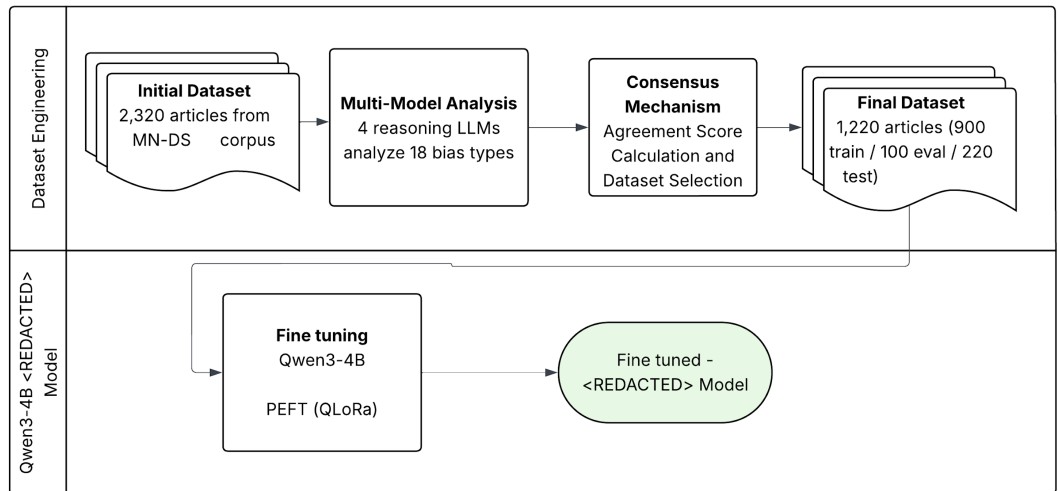

Figure 1: <**Redacted**>methodology overview showing the two main phases: dataset engineering with consensus-based validation and model fine-tuning using Parameter-Efficient Fine-Tuning techniques.

Four reasoning-capable Large Language Models were employed to analyze each article: Claude 3.7 (Anthropic Team, 2025), DeepSeek-R1 (DeepSeek-AI et al., 2025), o3-mini (OpenAI, 2025), and Gemini 2.5 (Google Team, 2025). Each model received identical prompts to identify and classify 18 distinct bias types across four granularity levels: None (0), Low (1), Moderate (2), and High (3).

### 2.1.1 AGREEMENT SCORE CALCULATION

To ensure dataset quality, we implemented a distance-based agreement scoring mechanism. First, we mapped qualitative bias levels to numerical values as shown in Table 1.

Table 1: Bias level mapping to numerical values

| Bias Level | Numerical Value |
|:----------:|:---------------:|
| None | 0 |
| Low | 1 |
| Moderate | 2 |
| High | 3 |

The agreement score $A$ between two models $i$ and $j$ for a specific bias type is calculated using the following formula:

$$A(i,j) = \begin{cases} 1.0 & \text{if } L_i = L_j \\ 0.75 & \text{if } L_i > 0, L_j > 0 \text{ and } |L_i - L_j| = 1 \\ 0.5 & \text{if } L_i > 0, L_j > 0 \text{ and } |L_i - L_j| = 2 \\ 0.0 & \text{if } (L_i = 0 \text{ and } L_j \neq 0) \text{ or } (L_i \neq 0 \text{ and } L_j = 0) \end{cases} \quad (1)$$

where $L_i$ and $L_j$ represent the numerical bias levels assigned by models $i$ and $j$ respectively. This relationship is also illustrated in the agreement matrix shown in Appendix B.

This scoring system assigns perfect agreement (1.00) for identical classifications, high agreement (0.75) for adjacent bias levels within the bias spectrum, moderate agreement (0.50) for bias levels

differing by two steps, and zero agreement (0.00) when one model detects no bias while another detects any level of bias.

The overall agreement score for each article is computed by averaging all pairwise agreement scores across models and bias types:

$$S_{overall} = \frac{1}{18} \sum_{b=1}^{18} \frac{1}{6} \sum_{\text{all pairs}} A(M_i, M_j)_b \qquad (2)$$

where $b$ represents each of the 18 bias types, the sum covers all six possible pairwise combinations of the four models, and $A(M_i, M_j)_b$ is the agreement score between models $i$ and $j$ for bias type $b$.

### 2.1.2 DATASET SELECTION PROCESS

Our dataset selection procedure involved two sequential filtering steps:

**Step 1 - Statistical Outlier Removal**: We removed articles with agreement scores below -2 standard deviations from the mean.

**Step 2 - Bias Confirmation Requirement**: We implemented a confirmation mechanism using Claude 3.7 as our baseline model (selection rationale detailed in 3.1.1). For each bias type present in an article, we required that at least one additional model must confirm the baseline model's bias classification to ensure reliability.

The confirmation rule was defined as:

- If Claude 3.7 classifies a bias as present (level $> 0$), at least one of the three remaining models must also classify it as present
- If Claude 3.7 classifies a bias as absent (level $= 0$), at least one of the three remaining models must also classify it as absent
- Articles where Claude 3.7's classification cannot be confirmed by any other model for any bias type are excluded from the final dataset

This baseline-confirmation approach ensures that bias detection is not based on single-model predictions, thereby reducing hallucinations and improving dataset reliability. The filtering process resulted in a utilization rate of 1,220 out of 2,320 articles (52.6%), significantly improving dataset quality by ensuring multi-model consensus on bias classifications.

### 2.1.3 FINAL DATASET COMPOSITION

The cleaned dataset was partitioned into 900 articles for training (73.8%), 100 for validation (8.2%), and 220 for testing (18.0%).

## 2.2 MODEL FINE-TUNING

We fine-tuned the Qwen3 4B model (Yang et al., 2025) using Parameter-Efficient Fine-Tuning (PEFT) techniques, specifically employing QLoRA (Quantized Low-Rank Adaptation) (Dettmers et al., 2023). The training process utilized the 900-article training set, with the 100-article validation set employed for checkpoint selection. Full hyperparameters are provided in Appendix C.

**Evaluation Metrics**: Model performance was evaluated based on agreement scores with Claude 3.7 and JSON output validity. JSON validity is crucial as our models are required to produce structured outputs following a specific schema for bias classification. Invalid JSON outputs indicate parsing failures that render the analysis unusable, making this a critical reliability metric alongside agreement scores.

The fine-tuning process was designed to transfer the reasoning and bias detection capabilities demonstrated by the larger models to the more computationally efficient Qwen3 4B architecture, following the "Less-Is-More" principle for reasoning (Ye et al., 2025). This approach enables the model to generate transparent, step-by-step reasoning processes while maintaining high performance in bias classification tasks.

# 3 RESULTS

Our results are presented in two main parts: dataset engineering outcomes demonstrating the effectiveness of our multi-model consensus approach, and fine-tuning results showing the performance of the <**Redacted**>model.

## 3.1 DATASET ENGINEERING RESULTS

The dataset engineering phase involved comprehensive analysis of model agreement patterns and validation of our consensus-based filtering approach. We present both the multi-model agreement analysis and the effectiveness of our data selection procedures.

### 3.1.1 MULTI-MODEL AGREEMENT ANALYSIS

We conducted a comprehensive agreement analysis across all four reasoning-capable LLMs using the complete dataset of 2,320 articles before applying our cleaning procedures. The pairwise agreement scores between models are presented in Table 2.

Table 2: Pairwise agreement scores between LLM models. The highest scores are indicated in bold.

| Model Pair | Claude | DeepSeek-R1 | Gemini 2.5 | o3-mini |
|---|---|---|---|---|
| **Claude** | - | 0.803 | **0.827** | 0.791 |
| **DeepSeek-R1** | 0.803 | - | 0.804 | 0.784 |
| **Gemini 2.5** | **0.827** | 0.804 | - | 0.808 |
| **o3-mini** | 0.791 | 0.784 | 0.808 | - |

The analysis reveals that Claude and Gemini 2.5 achieved the highest pairwise agreement score of 0.827, indicating strong consensus in their bias assessments. The second highest agreement was observed between Gemini 2.5 and o3-mini (0.808), further reinforcing Gemini 2.5's position as the most consistently aligned model across different reasoning approaches. To evaluate overall model coherence, we calculated each model's average agreement with all other models, as shown in Table 3.

Table 3: Model coherence scores (average agreement with other models). Selected baseline model is highlighted.

| Model | Coherence Score |
|---|---|
| **Claude 3.7** | **0.807** |
| DeepSeek-R1 | 0.797 |
| Gemini 2.5 | 0.813 |
| o3-mini | 0.794 |

While Gemini 2.5 demonstrated the highest overall coherence score of 0.813, we selected Claude 3.7 as our baseline model for the consensus mechanism due to a critical requirement: our methodology requires explicit reasoning text to train the Qwen3 4B model to generate transparent, step-by-step bias analysis. At the time of our experiments, only Claude 3.7 and DeepSeek-R1 provided detailed reasoning tokens through their API responses, while Gemini 2.5 and o3-mini did not expose their internal reasoning processes. Claude 3.7's strong coherence score of 0.807, combined with its essential reasoning outputs, made it the optimal choice for training the <**Redacted**>model to articulate bias detection decisions with the transparency and interpretability that are core objectives of our approach.

### 3.1.2 DATASET CLEANING EFFECTIVENESS

The two-step filtering process successfully reduced the dataset from 2,320 to 1,220 articles (52.6% utilization rate) while significantly improving data quality. The statistical outlier removal (Step 1)

eliminated articles with agreement scores below the -2 standard deviation threshold. The confirmation requirement (Step 2) removed articles where at least one bias type lacked multi-model confirmation, ensuring that all retained data points represent genuine consensus among the reasoning models.

## 3.2 FINE-TUNING RESULTS

We evaluated our fine-tuned model, denoted **Qwen3/4B <Redacted>**, against baseline Qwen3 models across multiple dimensions. Table 4 presents a comprehensive comparison of model performance on the test set of 220 articles.

Table 4: Comprehensive model performance comparison. The highest scores are indicated in bold.

| Model | Avg. Claude Agreement | Invalid JSON (%) | Thinking Length (words) | Claude Thinking Length (words) |
|---|---|---|---|---|
| **Qwen3/4B <Redacted>** | **0.8459** | 13 (5.91%) | 8009 ± 1750 | 9344 ± 2575 |
| Qwen3/4B | 0.7505 | 49 (22.27%) | 5478 ± 1247 | 9344 ± 2575 |
| Qwen3/32B | 0.8004 | 0 (0%) | 5008 ± 1048 | 9344 ± 2575 |

To ensure fair comparison, we conducted an additional evaluation using only the 207 articles where both models produced valid JSON outputs. Table 5 shows the head-to-head performance between our fine-tuned model and the larger Qwen3/32B baseline.

Table 5: Direct comparison on articles with valid JSON outputs (207 out of 220 test articles). The highest scores are indicated in bold.

| Model | Avg. Claude Agreement |
|---|---|
| **Qwen3/4B <Redacted>** | **0.8459** |
| Qwen3/32B | 0.7961 |

This controlled comparison confirms that our fine-tuned 4B model consistently outperforms the 32B baseline by 6.3% (0.8459 vs 0.7961), demonstrating that the "Less-Is-More" principle combined with high-quality reasoning examples can enable smaller models to achieve superior performance compared to larger, non-specialized models.

## 3.3 BIAS PATTERN ANALYSIS

Figure 2 provides a detailed visualization of pairwise agreement patterns across all 18 bias categories, including our fine-tuned model. The radar chart reveals that models achieve consistently high agreement across most bias types, with particularly strong consensus on demographic biases (political, gender, ethnic/cultural) and structural biases (statement bias, opinion-as-fact). The chart demonstrates that our **Qwen3/4B <Redacted>** model achieves agreement patterns very similar to Claude across all bias categories, indicating successful knowledge transfer. Some variation is observed in more subjective categories such as slant and source selection bias, reflecting the inherent complexity of these bias types.

The comprehensive agreement analysis across bias categories is further illustrated in Figure 3, which shows both pairwise and one-vs-others agreement statistics for all models including the fine-tuned versions. The analysis reveals that demographic bias categories achieve the highest agreement scores, with maximum pairwise agreement consistently above 0.95. More complex linguistic biases show greater variability, reflecting their nuanced nature. The inclusion of our fine-tuned model maintains the overall agreement patterns observed in the original dataset, with the **Qwen3/4B <Redacted>** model contributing positively to the consensus.

To understand model behavior patterns, we analyzed the sensitivity of each model to different bias categories, as shown in Figure 4. This analysis reveals distinct detection patterns: Claude and Gemini 2.5 show higher sensitivity to political and opinion-based biases, while DeepSeek-R1 and o3-mini demonstrate more conservative detection patterns. Our **Qwen3/4B <Redacted>** model exhibits sensitivity patterns closely aligned with Claude, further confirming successful knowledge transfer.

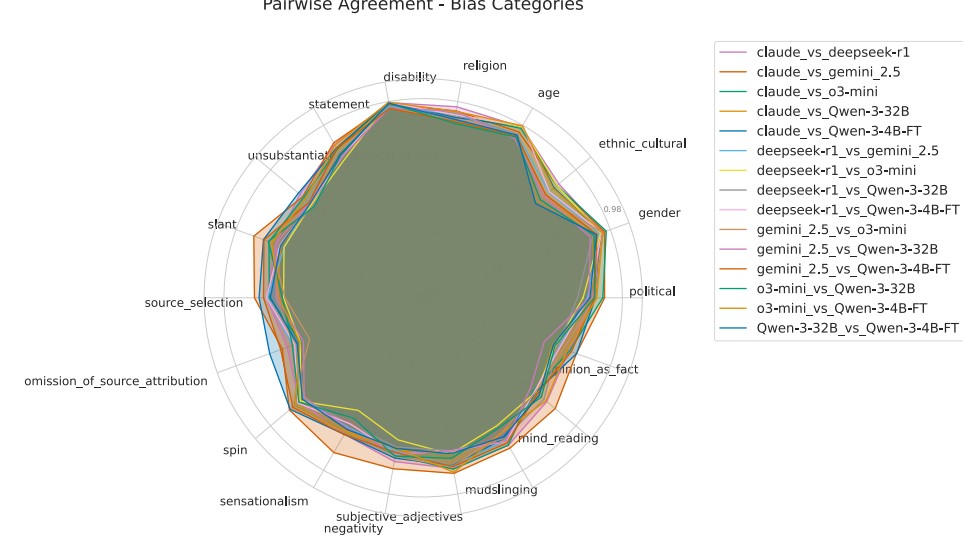

Figure 2: Pairwise agreement across 18 bias categories for four reasoning LLMs. The radar chart shows agreement scores (0–1) with lines for pairwise comparisons among Claude, DeepSeek-R1, Gemini 2.5, and o3-mini, and demonstrates knowledge transfer to the fine-tuned Qwen3-4B.

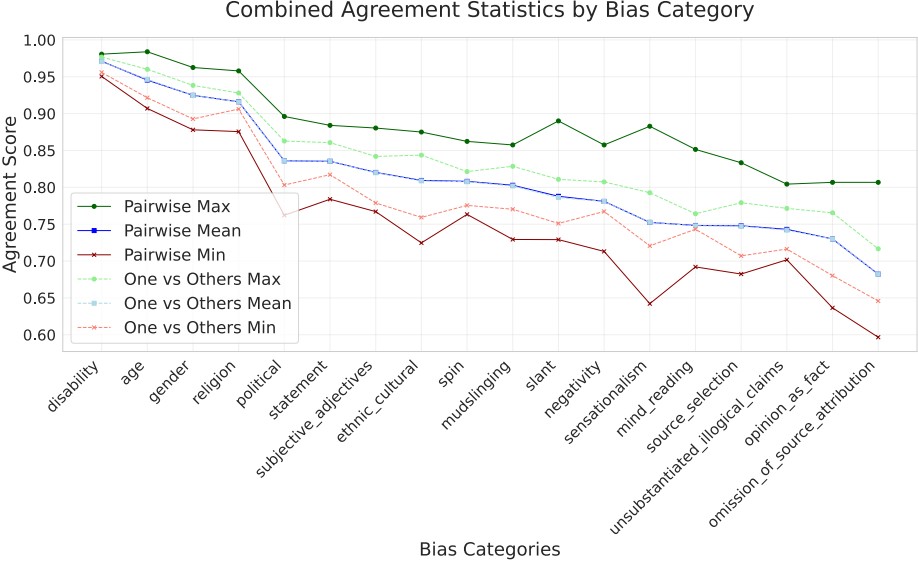

Figure 3: Combined agreement statistics by Bias category for the original four LLMs showing pairwise and one-vs-others agreement patterns.

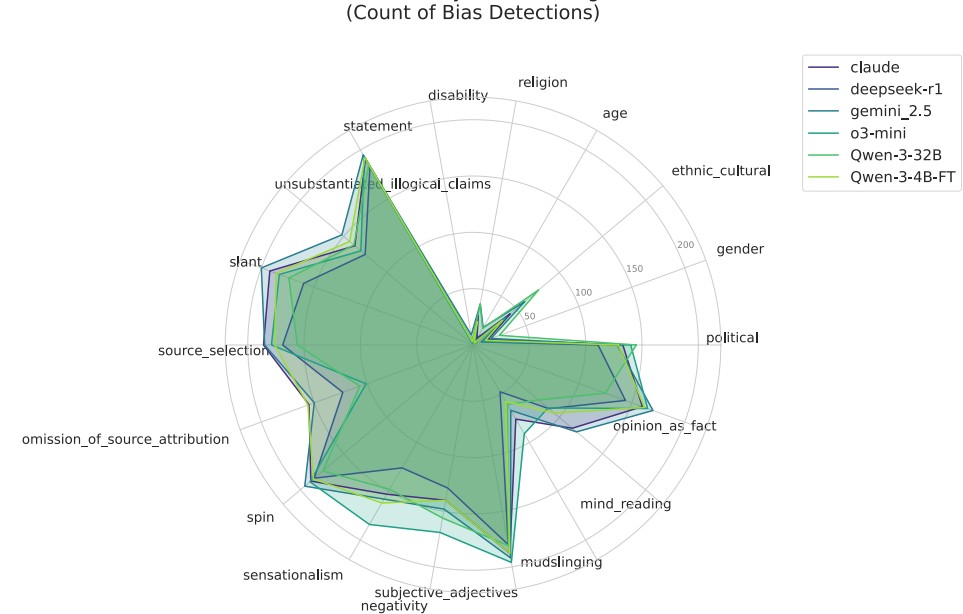

Figure 4: Model sensitivity to bias categories measured by detection frequency, including fine-tuned Qwen model, revealing distinct patterns across models.

### 3.4 KEY ACHIEVEMENTS

The fine-tuning results validate several core aspects of our approach. Most notably, our model achieved the highest agreement score (0.8459) with Claude among all tested models, indicating successful knowledge transfer from the reasoning examples. The approach demonstrated remarkable computational efficiency, showing that a 4B parameter model can outperform a 32B parameter model through targeted fine-tuning—representing an 8x reduction in model size while maintaining superior performance.

The model generates comprehensive reasoning outputs (8009 ± 1750 words) that approach the depth and detail of Claude's original reasoning, enabling transparency and interpretability. Additionally, it achieved significantly improved output reliability with a low invalid JSON rate (5.91%) compared to the baseline 4B model (22.27%), indicating enhanced structural consistency. The high agreement with Claude confirms that the model successfully learned to replicate the reasoning patterns and bias detection capabilities demonstrated in the training examples.

These results demonstrate that our consensus-based fine-tuning approach successfully creates a compact, efficient model that maintains the analytical depth and transparency of larger reasoning models while requiring significantly fewer computational resources.

## 4 CONCLUSION

This research demonstrates that the "Less-Is-More" principle can be effectively applied to media bias detection through a novel consensus-based fine-tuning approach. By combining multi-model agreement validation with targeted knowledge distillation, we address the critical challenge of building transparent, efficient, and reliable bias detection systems.

### 4.1 KEY CONTRIBUTIONS AND IMPLICATIONS

Our study makes several significant contributions:

**Consensus-based dataset construction:** We introduced a generalizable framework for constructing high-quality training data using multiple reasoning-capable LLMs. The framework defines a novel

distance-based agreement scoring function with statistical filtering and baseline confirmation. This approach is broadly applicable to tasks beyond media bias where multiple weak annotators produce structured outputs.

**Bias dataset and taxonomy:** Applying this framework, we curated the first open-source dataset of media bias annotations enriched with explicit reasoning traces across 18 bias types. The dataset, comprising 1,220 articles with structured JSON labels, will be released publicly to encourage reproducibility and further research.

**Efficient reasoning transfer:** We demonstrated that targeted fine-tuning of a 4B parameter model (Qwen3/4B) using QLoRA achieves +6.3% accuracy over a 32B baseline while generating interpretable reasoning outputs. This validates the "Less-Is-More" hypothesis that smaller, specialized models can outperform larger general-purpose systems when trained on carefully curated data.

**Transparency and interpretability:** By leveraging explicit reasoning outputs, our <**Redacted**>model produces detailed analytical explanations that allow users to understand and validate model decisions. This addresses critical needs for journalistic transparency and reliability in production systems.

**Robust dataset engineering:** Through a two-step filtering process, we achieved a 52.6% utilization rate with multi-model confirmation, ensuring consistency, diversity, and structured outputs suitable for downstream automation.

**Open-source release:** To foster transparency and reproducibility, we will release the full codebase, curated dataset, and fine-tuned model weights under an open license (assets will be made public upon acceptance).

## 4.2 LIMITATIONS AND FUTURE WORK

While our results are promising, limitations remain. Our evaluation focused on English-language news articles, leaving open questions about generalization across languages and cultural contexts. The approach depends on API-based reasoning models, which may restrict reproducibility. Future directions include exploring synthetic reasoning generation, extending to open-source reasoning-capable models, and adapting the taxonomy to capture emerging bias types.

## 4.3 FINAL REMARKS

In summary, our work provides a foundation for consensus-driven dataset construction, efficient reasoning transfer, and interpretable bias detection. The <**Redacted**>model highlights that smaller models, when trained on consensus-validated data, can achieve state-of-the-art performance while remaining computationally efficient. This methodology opens new opportunities for democratizing access to trustworthy bias analysis tools and lays the groundwork for advancing interpretable and scalable consensus-based learning across domains.

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

# A    COMPREHENSIVE BIAS TYPE TAXONOMY

This appendix provides a complete reference of the 18 bias types used in our <**Redacted**>model. The taxonomy integrates classifications from AllSides (Mastrine et al., 2019), Rodrigo-Ginés et al. (2024), Raza et al. (2022), and other established sources (Spinde et al., 2021; Hamborg et al., 2019). Each bias type is defined with examples and source attributions to ensure comprehensive coverage of media bias manifestations in news content.

Table 6: Complete taxonomy of 18 bias types with definitions and examples

| ID | Bias Type | Description |
|----|-----------|-------------|
| 1 | Political | Favors or criticizes a specific political viewpoint. *Example:* "The radical left continues to sabotage the economy." |
| 2 | Gender | Reinforces stereotypes or prejudices based on gender. *Example:* "The female engineer surprisingly solved the problem." |
| 3 | Cultural / Ethnicity | Unfairly portrays or generalizes ethnic or cultural groups. *Example:* "Immigrants are taking away local jobs." |
| 4 | Age | Unfairly stereotypes or discriminates based on age. *Example:* "Older employees rarely adapt to new technology." |
| 5 | Religion | Unfairly stereotypes or discriminates based on religion. *Example:* "Muslim neighborhoods are often hotspots of radicalism." |
| | | Continued on next page |

| ID | Bias Type | Description |
|---|---|---|
| 6 | Disability | Portrays individuals with disabilities or mental health conditions in a negative, stereotypical, or dehumanizing way. Often includes outdated, offensive language or implies that disability or mental illness is shameful, dangerous, or abnormal. *Example:* "This facility is for retarded individuals." |
| 7 | Statement Bias (labelling and word choice) | Also called presentation bias, refers to how articles choose to inform about certain entities/concepts through loaded language or presenting one side as the only side. Labelling uses specific words to convey particular opinions. *Example:* Words like "gender-affirming care" vs. "sex reassignment procedure" or "racial justice protest" vs. "riot" reveal different perspectives on the same events. |
| 8 | Unsubstantiated or Illogical Claims | Occurs when journalists make claims without supporting evidence or use flawed reasoning to reach unjustified conclusions. Includes both unsubstantiated claims and logical fallacies. *Examples:* "The senator's absence clearly shows he doesn't care about the crisis" (no source + unjustified inference); "This political change caused an increase in crime" (false cause fallacy). |
| 9 | Slant (Bias by Omission) | Highlights or plays up one particular angle while ignoring other perspectives. Through cherry-picking information, slant prevents readers from getting the full story and narrows understanding scope. |
| 10 | Source Selection Bias | The tendency to choose sources that support the story rather than sources that provide accurate accounts. *Example:* Covering an environmental disaster by only interviewing company representatives without giving voice to affected community members or independent experts. |
| 11 | Omission of Source Attribution | Occurs when journalists don't back up claims with sources, or sources are diffuse or unspecific. Sometimes intentional to protect source anonymity. *Examples:* Phrases like "according to a source," "critics say," or "experts believe." |
| 12 | Spin | Occurs when journalists try to create a "memorable story" using loaded or emotional language, exaggeration, or selective fact presentation to make content more interesting. Includes "clickbait" headlines and drama-focused stories. |
| 13 | Sensationalism | Information is exaggerated to create emotional reactions, targeting and provoking readers' emotions. Often involves selective information that supports certain views while omitting contradictory information. *Example:* "Bloodbath at the debate stage last night!" |
| 14 | Negativity Bias | Emphasizes bad or negative news, or frames events negatively. Follows the media adage "If it bleeds, it leads." *Example:* "The country is collapsing under the weight of failed leadership." |
| 15 | Subjective Adjectives | Uses qualifying adjectives that characterize or attribute specific properties to nouns, suggesting how readers should interpret issues rather than presenting facts objectively. *Example:* "The disturbing trend in education continues" or "The politician made a serious allegation." |
| 16 | Ad Hominem / Mudslinging | Makes unfair or insulting accusations to damage someone's reputation, or attacks a person's character instead of addressing their arguments or ideas. *Example:* "He's a clown with no experience or credibility." |
| | | Continued on next page |

| ID | Bias Type | Description |
|----|-----------|-------------|
| 17 | Mind Reading | Assumes knowledge of what another person thinks, interpreting internal thoughts or emotions of individuals who haven't explicitly expressed such thoughts or feelings. *Example:* "She clearly intended to undermine the election." |
| 18 | Opinion-as-Fact | Uses subjective language or statements under the guise of objective reporting. Presents subjective statements as factual information in supposedly objective news pieces. *Example:* "This policy is proof that the government doesn't care about citizens." |

This comprehensive taxonomy serves as the foundation for our multi-model consensus approach, enabling systematic identification and classification of bias across diverse news content. Each bias type is evaluated at four granularity levels (None, Low, Moderate, High) to provide nuanced analysis of bias intensity and manifestation patterns in news articles.

## B AGREEMENT SCORE MATRIX

Table 7: Agreement score matrix between bias level assessments

| Model 1 \ Model 2 | None | Low | Moderate | High |
|-------------------|------|-----|----------|------|
| None | 1.00 | 0.00 | 0.00 | 0.00 |
| Low | 0.00 | 1.00 | 0.75 | 0.50 |
| Moderate | 0.00 | 0.75 | 1.00 | 0.75 |
| High | 0.00 | 0.50 | 0.75 | 1.00 |

## C FINE-TUNING HYPERPARAMETERS

We fine-tuned Qwen3 4B using QLoRA with the following settings: LoRA rank 32, LoRA $\alpha$ 64, dropout 0.05, AdamW optimizer with learning rate 2e-4, batch size 64 with gradient accumulation of 8, trained for 4 epochs. We applied weight decay 0.01 and a warmup ratio of 0.1.

## D LARGE LANGUAGE MODEL (LLM) USAGE

In accordance with common disclosure practices for the use of large language models (LLMs) in scientific writing, we report the following:

- **Writing and polishing:** LLMs were used to aid in improving clarity, grammar, and conciseness of the manuscript text. All technical content, experimental design, and claims were authored and verified by the authors.

- **Retrieval and discovery:** LLMs were used to assist in identifying related work and refining literature searches. All references included in the paper were manually checked and validated by the authors.

- **No ideation or analysis:** LLMs were not used to generate novel research ideas, design experiments, or analyze results. All methodological and experimental contributions are original work by the authors.

- **Transparency:** Where LLMs were employed (e.g., text editing and discovery of related work), their contributions were limited to supportive tasks and do not constitute authorship.

