# OpenReview forum: "<REDACTED>: APPLYING THE LESS-IS-MORE PRINCIPLE TO MEDIA BIAS DETECTION IN NEWS ARTICLES"
_ICLR.cc/2026/Conference — Submitted to ICLR 2026_

### Official Review · Reviewer_FG9H · 2025-10-22

**Soundness:** 1
**Presentation:** 3
**Contribution:** 2
**Rating:** 2
**Confidence:** 4

**Summary:**

The paper presents a method for producing a multi-label bias dataset using consensus among LLMs labeling 18 different types of bias and then uses that dataset to fine-tune adapters on a (relatively) small LLM to achieve structured bias labeling with reasoning traces. The paper then applies these two methodologies to a news dataset labeled for topics and sub-topics to evaluate the methodologies. The paper finds that it can effectively create a bias dataset and model with relatively few, high-quality examples (on the order of thousands).

**Strengths:**

The paper is attacking a significant problem and proposes novel and potentially very useful combinations of existing LLM-based methods, namely consensus of LLM outputs, reasoning, and fine-tuning adapters. The idea of characterizing the bias present in news or blog articles is of significant societal value. And, the proposed method in the paper also incorporates LLM reasoning as part of the solution, which overcomes the limitations of more black box methods of characterizing bias.

**Weaknesses:**

The paper does have some issues with its validity and novelty claims. For validity, the empirical validation does not really cement the quality of the proposed methods. The evaluation metrics only look at between model agreement. While this is certainly a signal that the bias characterizations are correct, they are not the ground truth. For example, while unlikely, all of the models could be wrong in their bias classifications and agree on those erroneous bias classifications, which would lead to wrong conclusions about the correctness of the method. The method really needs to be compared to benchmark, bias-labeled datasets, or otherwise have the created dataset’s labels confirmed. I think it would be especially interesting to see if one could fine-tune a model in the proposed method and dataset and then apply it to a different bias benchmark that likely has slight differences in the bias labeling (i.e., a cross-dataset test as is done with related concepts like stance in Ng and Carley “Is my stance the same as your stance? A cross validation study of stance detection datasets.”). Such a test would really establish the ability of the method to correctly characterize bias in an open-world setting and handle the uncertainty and nuance is the various types of bias that exist.

In terms of novelty, the idea of using a consensus of LLMs to produce a labeled dataset is not a novel idea. For example, Cruickshank et al. Do this in “DIVERSE: A Dataset of YouTube Video Comment Stances with a Data Programming Model. I suspect the consensus function is novel, but that means the claims of novelty need to be scoped to this.

Finally, I would argue the paper shows more of the distillation principle than the less-is-more principle. The idea of fine-tuning a smaller model based on larger model outputs and having that smaller model even outperform larger models on the in-domain task is really more a demonstration of distillation.

**Questions:**

I have a number of questions around the fine-tuning procedure that need to be addressed to improve the paper.
    • How were the examples posed to the LLM during the training stage? It would be great to have an example of the formatted training prompt in one of the appendices.

    • What was the training methodology for the adapters? Was this supervised fine-tuning? And if, so, how did you calculate the loss? For example, did you use exact token prediction for the loss (i.e., if “moderate” bias was presen,t then the LLM would have to output the exact tokens of “moderate” to be correct)?

    • Can you give an example of a reasoning output from the fine-tuned model? It would be interesting to see if the reasoning, at least anecdotally, actually relates to the quality of the bias classifications and could serve as a guide for understanding bias in articles for humans.

    • Did you consider having o3 and Gemini 2.5 give a reasoning for their bias scores? While I fully understand that you cannot get the reasoning traces from the APIs, you could still ask the model to explain its reasoning for its score, so as to get some kind of reasoning traces.

    • Is it possible to use the reasoning traces as part of the consensus process to score bias? For example, a common pattern used especially in Agents and in analogous works like Lan et al. “Stance detection with collaborative role-infused llm-based agents”, which attacked the problem of stance classification, is to have an evaluator prompt or agent look at the answers and reasoning given by the ensemble of other models to produce the final output.

---

### Official Review · Reviewer_QxgA · 2025-10-30

**Soundness:** 3
**Presentation:** 3
**Contribution:** 1
**Rating:** 2
**Confidence:** 3

**Summary:**

This paper describes an approach to media bias detection that utilises an ensemble of LLMs to label articles with levels of bias displayed and then fine tunes a relatively modest LLM (Qwen3 4B) for bias detection using this dataset. The fine tuned model is shown to perform well at bias detection (based on the generated bias labels), and better than a larger model (Qwen3 32B).

**Strengths:**

The main strengths of the paper are:

-- The paper addresses an important area as media bias can lead to significant and negative societal impacts.
-- The paper is clearly organised and well written.
-- The experiments are well designed.
-- There are interesting ideas the ensemble based approach to label generation.
-- the authors plan to make their code and datasets publicly available.

**Weaknesses:**

The main weaknesses of the paper are:

-- The overall contribution is not completely obvious. The dataset used is synthetically labelled and the fine tuned model is shown to be capable of learning these labels. However, without ground truth labels the value of this for bias detection is not obvious.
-- The authors make claims about “Transparency and interpretability“ but never support these claims in the paper.
-- No details of prompts used for label generation are given.
-- The authors do not justify the need for a new bias taxonomy - what is missing in existing ones? Moreover, the combined bias taxonomy seems to combine two different kinds of things - one is a set of targets of bias (i.e. age, gender etc) while the other is a list of how different biases manifest. No justification for combining these into a single taxonomy is given.
-- The coverage of the literature on automated media bias detection, which is rich, given int eh paper is very limited.

**Questions:**

-- Why is the agreement set to 0 when L1 = 0 and L2 =1 L2 = 1? It seems excessive to implement such a severe score. Surely a pair of None and Low should have a higher score than a pair of None and High?

-- "thereby reducing hallucinations “ It is not obvious how hallucinations arise, or would need to be reduced, in this simple one-shot scenario. Why is this important?

-- "significantly improving dataset quality by ensuring multi-model consensus on bias classifications.”  What is dataset quality improved upon? This is a synthetic labelling procedure and there are no other labels to compare these to.

-- What prompts were used for one-shot models?

-- Why is the combined taxonomy needed and why is this combination of different things valid?

---

### Official Review · Reviewer_a8ne · 2025-11-01

**Soundness:** 3
**Presentation:** 4
**Contribution:** 2
**Rating:** 6
**Confidence:** 4

**Summary:**

In this paper the author constructed an automatically labeled dataset for news article bias through a multi-step annotation and verification process by 4 SOTA LLMs and various agreement and validity metrics. Using the dataset they then fine-tuned an open source model (Qwen 3/4b) on the bias detection task, demonstrating increased performance over their baseline, a larger Qwen 3/32b model. The authors also discuss the overall prevalence of agreement across LLMs in the 18 proposed bias categories, drawing some conclusions on agreement with the labeling models.
The paper offers a novel unsupervised dataset, a fine-tuned model for news bias detection and conclusions on the performance of smaller fine-tuned models for the bias-detection task as major contributions.

**Strengths:**

Originality: The paper broadens the complexity of bias detection with their unsupervised annotation framework. The dataset as well as the methodology for its genesis has the potential to be of further use to research within the space of fine-tuning and knowledge-transfer and bias detection.

Quality: Experimental results on fine-tuning are decent and offer some insights into how well the model manages to align with the task. The dataset presented in the paper could provide an entry point into either the methodology of the bias category definition and further experimentation through extension of the data or evaluation on different model families.

Clarity: The paper follows a clearly understandable methodological approach in setting out the experiments. They are able to draw convincing conclusions and account for different aspects in bias detection (categories and bias level). Their conservative approach with regards to annotation agreement is a good basis for a focused dataset contribution.

Significance: There is a fair contribution in regards to providing a novel dataset, as well as demonstrating promising fine-tuning and alignment results.

**Weaknesses:**

There is no validation in the dataset beyond agreement with other automated LLMs. This places it firmly in the unsupervised annotation category. At the same time no studies have been provided on how well the LLMs that are used for this task are generally at detecting these types of bias in either news, or other settings. Some insights here could help alleviate the uncertainty of general mis-alignment of the source models.
While the intensity of bias categories were part of the annotation process there is no data on how the distribution of these categories look across the annotated examples.
The final score is computed entirely through agreement with Claude 3.7, disregarding the other models, but the choice for only going with Claude 3.7 is not substantiated enough.

**Questions:**

There is very little mention in the text on how exactly the provided dataset is fundamentally different from other existing datasets within this space. Can you elaborate more on what you would consider the distinguishing features of the dataset?

Can you draw some conclusions to the prevalence of indicated bias in articles? In some of the categories almost all articles seem to show these types of bias (statement, slant, spin etc) while others are basically not present at all (gender, age, disability). Would the trained model be able to pick up on the underrepresented categories at all, if we use a deliberately selected evaluation sample to check for this?

Figure 4: I found the different shades of green to be hard to distinguish in the graphic. Maybe rethink the chosen color palette.

---

### Meta-Review · Area_Chair_pNFN · 2025-12-22

**Summary:**

The paper constructs a news-bias dataset by prompting four LLMs to annotate articles across 18 bias categories, applying multi-step verification and agreement filtering. Using these synthetic labels, the authors fine-tune a 4B-parameter Qwen model and show it outperforms a larger Qwen 32B model on reproducing these same labels. They analyze inter-model agreement and discuss patterns in how often different bias categories are detected. The contribution claimed is a new dataset, a fine-tuned bias-detection model, and insights into LLM agreement on media bias.

Reviewers question the validity of the dataset, as all labels derive from LLM agreement rather than human or benchmark ground truth, making it unclear whether the dataset captures real bias. The novelty is overstated, since LLM-ensemble labeling is already common, and the taxonomy is introduced without justification. The evaluation only confirms that models can replicate synthetic labels, not that the system detects bias reliably in the real world. Missing prompt details, limited literature coverage, lack of cross-dataset testing, and no analysis of label distribution further weaken the contribution.

**Reviewer Concerns:**

No rebuttal from authors.

**Reviewer Scores:**

They won't change the scores.

---

### Decision · Program_Chairs · 2026-01-26

Reject